# Associations between Vitamin D, Omega 6:Omega 3 Ratio, and Biomarkers of Aging in Individuals Living with and without Chronic Pain

**DOI:** 10.3390/nu14020266

**Published:** 2022-01-09

**Authors:** Akemi T. Wijayabahu, Angela M. Mickle, Volker Mai, Cynthia Garvan, Toni L. Glover, Robert L. Cook, Jinying Zhao, Marianna K. Baum, Roger B. Fillingim, Kimberly T. Sibille

**Affiliations:** 1Department of Epidemiology, College of Public Health and Health Professions, and College of Medicine, University of Florida, Gainesville, FL 32610, USA; akemiwijayabahu@ufl.edu (A.T.W.); vmai@ufl.edu (V.M.); cookrl@ufl.edu (R.L.C.); jzhao66@ufl.edu (J.Z.); 2Emerging Pathogens Institute, University of Florida, Gainesville, FL 32610, USA; 3Department of Physical Medicine & Rehabilitation and Aging & Geriatric Research, University of Florida, Gainesville, FL 32610, USA; angela.mickle@ufl.edu; 4Pain Research and Intervention Center of Excellence, University of Florida, Gainesville, FL 32610, USA; RFillingim@dental.ufl.edu; 5Department of Anesthesiology, College of Medicine, University of Florida, Gainesville, FL 32610, USA; CGarvan@anest.ufl.edu; 6School of Nursing, Oakland University, Rochester, MI 48309, USA; tglover@oakland.edu; 7Robert Stempel College of Public Health and Social Work, Florida International University, Miami, FL 33174, USA; baumm@fiu.edu

**Keywords:** vitamin D, omega 6:omega 3 ratio, leukocyte telomere length, C-reactive protein

## Abstract

Elevated inflammatory cytokines and chronic pain are associated with shorter leukocyte telomere length (LTL), a measure of cellular aging. Micronutrients, such as 25-hydroxyvitamin D (vitamin D) and omega 3, have anti-inflammatory properties. Little is known regarding the relationships between vitamin D, omega 6:3 ratio, LTL, inflammation, and chronic pain. We investigate associations between vitamin D, omega 6:3 ratio, LTL, and C-reactive protein (CRP) in people living with/without chronic pain overall and stratified by chronic pain status. A cross-sectional analysis of 402 individuals (63% women, 79.5% with chronic pain) was completed. Demographic and health information was collected. Chronic pain was assessed as pain experienced for at least three months. LTL was measured in genomic DNA isolated from blood leukocytes, and micronutrients and CRP were measured in serum samples. Data were analyzed with general linear regression. Although an association between the continuous micronutrients and LTL was not observed, a positive association between omega 6:3 ratio and CRP was detected. In individuals with chronic pain, based on clinical categories, significant associations between vitamin D, omega 6:3 ratio, and CRP were observed. Findings highlight the complex relationships between anti-inflammatory micronutrients, inflammation, cellular aging, and chronic pain.

## 1. Background

Leukocyte telomere length (LTL) is widely studied as a biomarker of cellular aging. Telomere shortening has been linked to cellular aging processes, such as genomic instability and cellular senescence [1]. Not only does inflammation within the cellular environment contribute to telomere shortening, but the accumulation of senescent cells and the senescence-associated secretion of pro-inflammatory cytokines also contribute to the low-grade chronic inflammatory state of aging adults [1]. The elevated inflammatory state serves as a contributing factor in the pathogenesis and progression of many age-related disease conditions, including the development of chronic pain [2,3,4]. Shorter LTL is also associated with an increased risk of morbidity and mortality [5,6,7].

Chronic pain is defined as any persistent pain that lasts for more than three months, a debilitating condition with increased prevalence in older adults [8]. Chronic pain is associated with increased risk of age-related health outcomes, including mortality [8,9,10]. Previous studies have shown that greater chronic pain severity is associated with elevated levels of pro-inflammatory cytokines, such as C-reactive protein (CRP) [2,11,12,13] and shorter LTL [14,15,16,17]. The pathogenesis and progression of chronic pain appears to be influenced by a combination of physiological, psychosocial, and lifestyle factors [2,8,18]. Some of these factors, including nutritional patterns, are amenable to behavioral and clinical interventions [8].

Previous studies have suggested that individuals with chronic pain may benefit from consuming foods and micronutrients with anti-inflammatory properties [19,20,21,22]. However, the question remains if such food and micronutrients are associated with longer LTL in people living with chronic pain; 25-hydroxyvitamin D (vitamin D) and omega 3 fatty acids seem to show some benefit toward reducing chronic pain [19,23,24,25,26] and are associated with longer LTL [27,28,29]. Although previous studies indicate conflicting evidence of the association between vitamin D on LTL in the general population [30,31,32], two studies reported expression of LTL protective enzyme telomerase in individuals with chronic inflammatory conditions [28,33]. Moreover, there is a developing body of evidence showing vitamin D deficiency is positively associated with both inflammation and chronic pain, which may be related to changes in pain signaling pathways (e.g., endocannabinoids system) [34] and gut microbial composition [34,35,36].

Although omega 3 supplementation seems to show benefit towards reducing inflammation [37,38,39], the level of omega 6 could mask the anti-inflammatory activity of omega 3 [27,29]. While some studies show a positive association between omega 3 and LTL [40,41,42], others have shown null findings [43,44]. Thus, omega 6:omega 3 ratio (omega 6:3 ratio) may be more informative than omega 3 levels alone. In support of this approach, an omega 3 supplementation trial showed an inverse association between omega 6:3 ratio and LTL but no association with omega 3 alone [44]. Consistently, other studies have demonstrated that higher omega 6:3 ratio is associated with lower LTL [45] and higher levels of pro-inflammatory cytokines, such as CRP [46]. Importantly, Sibille et al. [47] reported that higher levels of omega 6:3 ratio (>5) were associated with greater clinical pain, functional limitations, experimental pain sensitivity, and psychosocial distress [47]. It remains unclear whether there is an association between omega 6:3 ratio and LTL in people living with chronic pain and if there might be an association between vitamin D and omega 6:3 ratio combined and LTL.

In the current study, we address the following aims by determining: (1) the independent and combined associations between vitamin D, omega 6:3 ratio, and LTL and (2) the independent and combined associations between vitamin D, omega 6:3 ratio, and CRP in people living with and without chronic pain. Micronutrients are analyzed as continuous variables and categorized based on clinical classifications. We hypothesized that higher levels of vitamin D (or sufficiency, > 30 ng/mL) and lower omega 6:3 ratio (or ≤ 5) are associated with longer LTL and lower levels of CRP in people living with and without chronic pain. Additionally, we anticipated that vitamin D and omega 6:3 ratio would show a stronger relationship with LTL and CRP than when considered individually.

## 2. Methods

### 2.1. Study Population and Design

This study was conducted using data from the Understanding Pain and Limitations in OsteoArthritic Disease (UPLOAD) cohort. The main goal of this cohort is to understand the complex array of biopsychosocial factors contributing to knee osteoarthritis and health disparities. Details of this cohort, including inclusion and exclusion criteria, have been previously described in detail [48,49]. Briefly, the UPLOAD cohort recruited individuals aged 45–85 years who are living with and without symptomatic knee pain [50]. Individuals with serious medical conditions (e.g., heart failure, history of myocardial infarction), peripheral neuropathy, systemic rheumatologic disorders, knee surgery, daily opioid use, cognitive performance with a score of ≤ 22 on the Mini-Mental Status Examination (MMSE), psychiatric hospitalization within the preceding year, and concern with study protocol, including experimental pain assessment and blood draws, were excluded. The UPLOAD study used both active and passive recruitment strategies, including posted fliers, radio and printed media advertisements, orthopedic clinic recruitment, and word-of-mouth referrals in recruiting participants from the two collaborating study sites at the University of Florida (UF) and the University of Alabama Birmingham (UAB). All study protocols utilized in the UPLOAD cohort were approved by the UF and the UAB Institutional Review Boards (IRBs). Individuals who agreed to participate provided written informed consent.

After enrollment, eligible study participants completed demographic and pain screening questions followed by a health assessment session to collect the following data: anthropometric and clinical pain measures, vital signs, health history, lifestyle factors, and current medication use. Within four weeks of the health assessment session, participants completed quantitative sensory testing, which included collection of vital signs and blood samples.

The current study is a cross-sectional analysis of data from the baseline UPLOAD cohort, collected between January 2010 and February 2014. The purpose of the current analysis is to evaluate the associations between selected micronutrients, vitamin D and omega 6:3 ratio, and biomarkers of aging. Participants with complete information for the predictors and outcomes were included in each analysis.

### 2.2. Measurements

#### 2.2.1. Serum Vitamin D (ng/mL)

Serum samples collected during the sensory testing session were stored in a −80 °C freezer until quantification of 25-hydroxyvitamin D (25(OH)D = serum vitamin D). The total serum level of vitamin D (25(OH)D = sum of 25(OH)D2 and 25(OH)D3) was quantified using High-Performance Liquid Chromatography (HPLC) within six months of sample collection. A detailed account of the sample processing and quantification has been previously published [49]. A vitamin D variable that accounted for seasonal and geographic variations was also tested. However, as there was no difference in the correlation analysis, we included the original unadjusted vitamin D variable. Additionally, we created a categorical vitamin D variable based on clinically meaningful categories: deficient (<20 ng/mL), insufficient (20–30 ng/mL), and sufficient (>30 ng/mL) [51].

#### 2.2.2. Serum Omega 3 and Omega 6 (ng/mL)

Serum samples collected during the sensory testing session were used to quantify omega 3 and omega 6. A detailed account of the sample processing and quantification has been previously published [47]. The serum level of α-Linoleic acid (ALA), Docosahexaenoic Acid (DHA), Eicosapentaenoic Acid (EPA), Arachidonic Acid (AA), and Linoleic Acid (LA) were quantified using Liquid Chromatography-Tandem Mass Spectrometry (LC-MS/MS). Total serum omega 3 was calculated by summing the levels of ALA, DHA, and EPA, and total omega 6 by summing AA and LA. The omega 6:3 ratio was computed using the total percent values of omega 6 and omega 3. Additionally, a categorical omega 6:3 ratio was created based on clinical recommendations [47,52], below or equal to the recommended ratio (≤5), and above the recommended ratio (>5).

#### 2.2.3. Leukocyte Telomere Length

Details of the blood extractions, LTL quantification, and quality control protocols have been published elsewhere [14]. Briefly, blood samples provided by participants were immediately processed to retrieve the buffy coat and then stored in a −80 °C refrigerator until DNA extraction. The LTL was analyzed using quantitative Polymerase Chain Reaction (qPCR) at the Blackburn lab, University of California San Francisco [53,54]. The relative telomere length (T/S ratio) was calculated by using the LTL and a single-copy gene (human ß-globulin gene = HBB) using protocols described by Cawthon et al. [53,54]. The HBB was used to assess the quality of gene amplification across study participants.

#### 2.2.4. C-Reactive Protein

Serum levels of CRP were quantified by analyte-specific Enzyme-Linked Immunosorbent Assay (ELISA) kit per standard laboratory protocol.

#### 2.2.5. Socio-Demographics

We included age (years), sex (women and men), ethnicity/race (non-Hispanic white and non-Hispanic black), highest level of education (≤high school, >high school), and annual household income ($0–$19,000, $20,000–$49,999, $50,000–$79,999, ≥$80,000).

#### 2.2.6. Lifestyle Factors

We included waist-hip ratio (WHR) defined using standard sex-specific cut-off values (low: women ≤ 0.80 and men ≤ 0.95, moderate: women 0.81–0.85 and men 0.96–1.00, high: women > 0.86 and men > 1.00) [55], physical activity (<1/week, 1–3 times/week, ≥4 times/week), and tobacco smoking status (never, former, and current smoker).

#### 2.2.7. Chronic Pain Status and Total Number of Pain Sites

Chronic pain was defined as pain for more days than not over the last 3 months at one or more pain sites. Total number of pain sites is a recognized measure of chronic pain severity [56]. The total number of pain sites was reported by participants on a self-reported questionnaire during the health assessment [57]. The total number of possible pain sites ranged from zero to 24. Chronic pain status was categorized as no chronic pain and chronic pain (≥1 chronic pain sites).

#### 2.2.8. Depressive Symptoms

Symptoms of depression were assessed using the Center for Epidemiology Studies Depression (CES-D) scale, which assesses depression symptoms during the past week. Data were categorized based on clinically defined cut-off values (No depression 0–9, mildly 10–15, moderate/severe ≥ 16) [58].

#### 2.2.9. Co-Morbidities

Self-reported current and past diagnoses of health conditions were summed to create a total number of comorbidities and categorized as no comorbidities, one comorbidity, and ≥2 comorbidities [57].

#### 2.2.10. Study Site

The two study sites, the University of Florida and the University of Alabama at Birmingham, were included as a covariate to account for any site-specific differences.

#### 2.2.11. Non-Steroidal Anti-Inflammatory Drug Use (NSAID)

The medication list was collected and reviewed by trained staff. NSAID use was defined as any NSAID use and no NSAID use.

### 2.3. Data Analysis

SAS version 9.4 was used for statistical analyses (SAS Institute Inc, Cary, NC, USA). To characterize the population, we used descriptive statistics that were expressed as mean ± standard deviation (SD) or as a percent of the population. Significance was defined as *p* < 0.05. Continuous variables were checked for normality. CRP and LTL were log-transformed.

#### 2.3.1. Correlation Analyses

We conducted Pearson correlation analyses to identify the associations between vitamin D, omega 6:3 ratio, LTL, and CRP.

#### 2.3.2. Regression Analyses

General linear models (“GLM” procedure) were used to estimate the strength and direction of associations. We conducted both univariable and multivariable models. We investigated the associations between micronutrients and outcomes of interest individually and in a combined model that included both vitamin D and omega 6:3 ratio. Micronutrient variables were treated as both continuous variables and as categorical variables based on clinical classifications.

#### 2.3.3. Covariate Selections for the Multivariable Regression Models

We considered sociodemographic factors (age, sex, ethnicity/race, highest level of education, and annual household income), lifestyle factors (WHR, physical activity, and tobacco smoking status), medical conditions (number of chronic pain sites, depressive symptoms, and number of comorbidities), and other (study site) for the multivariable models. We used multiple statistical methods to establish the robustness of variable selection (e.g., correlations and multicollinearity check, stepwise and penalized regression). The number of pain sites was included to account for the pain severity in the models.

Final models for aim 1 included age, sex, ethnicity/race, study site, WHR, physical activity, and number of pain sites in adjusted models. Final models for aim 2 included age, sex, annual household income, number of comorbidities, tobacco smoking status, WHR, physical activity, and number of pain sites.

### 2.4. Post-Hoc Analyses

We conducted regression analyses stratified by chronic pain status. Multivariable models were adjusted using previously selected covariates. To address possible anti-inflammatory medication influences, NSAID use was included as an additional covariate in the stratified analyses. Due to the sample size limitations of the no chronic pain group (LTL models *n* = 26 and CRP models *n* = 35), the regression analysis with categorical micronutrient variables were restricted to the chronic pain group only (LTL models *n* = 101 and CRP models *n* = 128).

## 3. Results

### 3.1. Participant Characteristics

Participant characteristics are summarized in Table 1. Briefly, participants had a mean age of 56.6 (SD 7.6) years; 63.4% were women, 52.0% were non-Hispanic white, 58.2% had an education beyond the high-school level, and 66.5% had an annual household income ≤ $49,999. Of the total population, 79.5% reported chronic pain. Of the 402 individuals in the study, for the LTL analysis, 209 had complete data for vitamin D, and 127 had complete data for omega 6:3 ratio. For the CRP analysis, 211 had complete data for vitamin D, and 163 had complete data for omega 6:3 ratio.

### 3.2. Micronutrients and LTL

#### 3.2.1. Associations between Vitamin D Continuous Variable and LTL

There were no statistically significant associations between the vitamin D and log-LTL. Unadjusted and adjusted regression results for vitamin D as a continuous variable are shown in Table 2.

#### 3.2.2. Associations between Vitamin D Clinical Categorical Variable and LTL

Statistically significant associations between vitamin D categories and log-LTL were not detected. Unadjusted and adjusted regression results for categorical vitamin D are shown in Appendix A.

#### 3.2.3. Associations between Omega 6:3 Ratio Continuous Variables and LTL

We did not find a statistically significant association between omega 6:3 ratio and log-LTL. Unadjusted and adjusted regression results for vitamin D as a continuous variable are shown in Table 2.

#### 3.2.4. Associations between Omega 6:3 Clinical Categorical Variable and LTL

Associations between omega 6:3 categories and LTL were not statistically significant Unadjusted and adjusted regression results for categorical omega 6:3 ratio are shown in Appendix A.

#### 3.2.5. Associations between Combined Vitamin D and Omega 6:3 Ratio and LTL

Statistically significant associations were not detected when micronutrient predictors were considered together. Results were consistent for both the continuous and categorically micronutrient variables. Regression results for combined continuous micronutrient models are shown in Table 2, and combined categorical micronutrient models are shown in Appendix A.

### 3.3. Micronutrients and CRP

#### 3.3.1. Associations between Vitamin D Continuous Variable and CRP

An inverse association between vitamin D and log-CRP (β = −0.04, 95% CI −0.05, −0.02) was detected for the unadjusted model. However, statistical significance was not maintained after adjusting for covariates. Unadjusted and adjusted regression results for vitamin D as a continuous variable are shown in Table 3.

#### 3.3.2. Associations between Vitamin D Clinical Categorical Variable and CRP

Vitamin D deficiency (<20 ng/mL) was positively associated with log-CRP (adjusted model: β = 0.56, 95% CI 0.10, 1.02, reference group: sufficiency > 30 ng/mL). Unadjusted and adjusted regression results for categorical vitamin D are shown in Appendix A.

#### 3.3.3. Associations between Omega 6:3 Ratio Continuous Variables and CRP

Omega 6:3 ratio and log-CRP (adjusted model: β = 0.11, 95% CI 0.01, 0.21) were positively associated. Unadjusted and adjusted regression results for vitamin omega 6:3 ratio as a continuous variable are shown in Table 3.

#### 3.3.4. Associations between Omega 6:3 Clinical Categorical Variable and CRP

Omega 6:3 ratio above the recommended limit (>5) was associated with higher levels of log-CRP (adjusted model: β = 0.75, 95% CI 0.12, 1.38, reference ratio: ≤ 5). Unadjusted and adjusted regression results for categorical omega 6:3 ratio are shown in Appendix A.

#### 3.3.5. Associations between Combined Vitamin D and Omega 6:3 Ratio and CRP

We did not find statistically significant association between continuous vitamin D, omega 6:3 ratio, and log-CRP. For categorical variables, the combined model was significant but only for the association between omega 6:3 ratio > 5 and log-CRP. The strength of the association was slightly increased after accounting for vitamin D status when compared to the individual omega 6:3 ratio model (β 0.76 vs. 0.75). Moreover, the combined micronutrient model showed better predictive value as opposed to individual micronutrient models (adjusted models: omega 6:3 ratio R^2^ = 0.39; vitamin D R^2^ = 0.37; combined R^2^ = 0.43). Regression results for combined continuous micronutrient models are shown in Table 3, and combined categorical micronutrient models are shown in Appendix A.

### 3.4. Post-Hoc Micronutrients and LTL Stratified by Pain Status

In a stratified group analysis by chronic pain status, an association between vitamin D, omega 6:3 ratio, and log-LTL was not observed.

### 3.5. Post-Hoc Micronutrients and CRP Stratified by Pain Status

There was a significant inverse association between vitamin D as a continuous variable and a positive association for vitamin D deficiency and log-CRP in the chronic pain group. Additional adjustment with NSAIDs did not change the association. There was a significant positive association between omega 6:3 ratio as a continuous and categorical variable (omega 6:3 ratio > 5) and log-CRP in the chronic pain group. Further adjustment with NSAIDs use showed a null association for the continuous omega 6:3 analysis and a slightly attenuated association for the categorical analysis.

In the chronic pain group, with combined vitamin D, omega 6:3 ratio, and log-CRP, we observed a linear association between omega 6:3 ratio and log-CRP independently to vitamin D; the association did not remain significant after accounting for NSAIDs use. However, the categorical analysis omega 6:3 ratio > 5 was significantly associated with log-CRP independent of vitamin D, even after adjusting for NSAIDs use. Multivariable regression analysis results for continuous micronutrients are shown in Appendix A, and those for categorical micronutrients are shown in Appendix A.

## 4. Discussion

The overall intention of this study was to investigate associations between serum vitamin D, omega 6:3 ratio, LTL, and CRP. There were no associations observed between vitamin D, omega 6:3 ratio, and LTL in independent and combined analyses. Although a relationship was not detected between continuous vitamin D and CRP, vitamin D deficiency was positively associated with CRP. Omega 6:3 ratio both as a continuous and a categorical variable was associated with higher levels of CRP. When vitamin D and omega 6:3 ratio were combined in the CRP analysis, a stronger model was observed. In individuals with chronic pain, based on clinical categories and with considerations for NSAID use, we observed significant associations between vitamin D, omega 6:3 ratio, and CRP. Findings highlight the complex relationships between anti-inflammatory micronutrients, inflammation, cellular aging, and chronic pain.

### 4.1. Vitamin D, Omega 6:3 Ratio, and LTL

We did not find an association between vitamin D, omega 6:3 ratio, and LTL. While some studies report a positive association between vitamin D and LTL [30,59,60,61,62], others report null findings [31,63,64,65]. Liu and colleagues reported ethnic/race group differences, noting a positive association in the group of white participants who had insufficient vitamin D [66]. Findings from a recent Mendelian randomization study did not support a causal association between genetically instrumented circulating vitamin D and LTL; however, the single-nucleotide polymorphisms (SNPs) used in instrumenting vitamin D levels were based on populations of European descent [67].

While the evidence linking the omega 6:3 ratio and LTL is limited [44], there is mixed evidence on the associations between omega 3 and LTL [40,41,42,43]. One of the randomized controlled trials reported no association between omega 3 and LTL and an inverse association between the omega 6:3 ratio and LTL [44]. Other studies have shown a positive association between omega 3 and LTL in people living with cardiovascular diseases [40,42] and cognitive impairment [41]. Importantly, LTL is a downstream measure of biological and psychosocial stress influenced by the cellular inflammatory milieu. Micronutrients are only one of numerous factors contributing to cellular aging, so not seeing strong relationships between the identified micronutrients and LTL may be due to the study sample size.

### 4.2. Vitamin D, Omega 6:3 Ratio, and CRP

The strength and the directionality of the associations between vitamin D and CRP appears to differ by conditions, and it is not always linear [68,69]. One study reported a strong inverse association between vitamin D and CRP in individuals with inflammatory diseases (acute and chronic) compared to non-inflammatory diseases (β −0.879 vs. −0.499) [69]. However, in those who have metabolic diseases, levels of CRP seem to drop faster with increasing vitamin D but slows after reaching vitamin D sufficiency [68]. In our study, we found a weak inverse association between serum vitamin D and CRP. However, when using clinical classification to categorize into deficiency (<20 ng/mL), insufficiency (20–30 ng/mL), and sufficiency (>30 ng/mL), we found that vitamin D deficiency is strongly associated with higher levels of CRP. Stratified analyses suggest that the relationship is specific to individuals with chronic pain.

Our stratified analysis indicated the associations observed between the omega 6:3 ratio and CRP were limited to those individuals living with chronic pain. Several studies have shown clinical benefits of having an omega 6:3 ratio between 2–5, with varying recommendations depending on the disease condition [52]. In general, an omega 6:3 ratio > 5 is implicated in negative health outcomes [52]. More importantly, higher levels of omega 6:3 ratio > 5 were associated with greater pain severity and functional limitations [47]. Thus, our findings highlight that micronutrient levels above the recommended range (>5) in individuals with chronic pain are associated with greater CRP levels than those who are below or equal to the recommended range (≤5), indicating greater systemic inflammation.

### 4.3. Biological Plausibility of Observed Associations

Vitamin D is involved in the regulation and expression of pro-inflammatory mediators, inflammatory genes and suppression of key inflammatory response pathways, such as NF-κB [70,71]. Possible relationships between vitamin D deficiency, pain signaling pathways (e.g., endocannabinoids system) [34], and altered gut microbial composition have also been indicated [34,35,36]. Similarly, omega 3 fatty acids are implicated in suppressing the inflammatory response, including the suppression of NF-κB activity, reducing the impact of omega 6 driven pro-inflammatory cytokine production [72]. There may also be some overlap between the anti-inflammatory activity of these micronutrients and pain reduction [37,73,74,75].

### 4.4. Limitations and Future Directions

There are several limitations to this study, one of which is the cross-sectional design; thus, the temporality of the associations cannot be determined. Second, the sample size might not be sufficient to capture the possible relationships between micronutrients and LTL. Third, the regression analysis with categorical micronutrients was restricted to the chronic pain group due to the small sample size of those without chronic pain. Finally, only 12 individuals had an omega 6:3 ratio in the recommended range (ratio ≤ 5). Thus, further investigations with more individuals in the recommended range would be helpful. Despite the limitations, findings from our study may provide an improved understanding of the potential relevance of clinically defined ranges of vitamin D and omega 6:3 ratio.

Regarding additional future directions, multiple time-point measurements might improve interpretations of inflammation. For instance, multiple time-point CRP measurements may provide a more accurate estimation of chronic systemic inflammation as opposed to using a single CRP measurement [76]. Although not specific to CRP, another study reported no statistically significant association between serum omega 3 and baseline LTL but found a significant positive association between serum omega 3 and change of LTL over time [40]. Even though some studies have shown cross-sectional associations between vitamin D, omega 3, and LTL [30,42,60], these examples indicate the possible relevance of using multiple time point measurements to assess both CRP and LTL measurements. Additionally, consideration of additional pro-inflammatory biomarkers, such as IL-6, may also be informative.

Further, to improve the robustness of covariate selection for multivariable regression models, we considered multiple different data-driven approaches. There may be unmeasured confounders and/or residual confounding. For example, alcohol intake was not available for our sample population. Although the associations between alcohol intake and outcomes of interest (CRP [77] and LTL [78,79]) are less clear, multiple studies have shown that heavy drinking may be associated with micronutrients, inflammation, and LTL [77,78,80,81,82,83]. Thus, considering alcohol use would be informative in future studies.

Finally, people living with chronic pain in our study population are representative of individuals with primarily musculoskeletal chronic pain. Thus, the relationship between micronutrients and CRP may differ by the type of pain experienced. For instance, while one study reported that vitamin D supplementation/maintaining sufficiency did not have a significant impact on CRP in individuals living with osteoarthritis [84], another study reported a significant reduction of CRP and surgical pain [85]. Therefore, our findings need to be replicated in individuals with differing pain conditions to better understand the relevance for chronic pain in general.

## 5. Conclusions

We observed overall patterns indicating relationships between vitamin D, omega 6:3 ratio, and CRP but not LTL. Findings suggest that the clinical categories of vitamin D and omega 6:3 ratio show a stronger relationship with CRP than when assessed as continuous variables. Specifically, vitamin D deficiency and higher levels of omega 6:3 ratio above the recommended level (>5) are associated with higher levels of CRP in people living with chronic pain. Further, the combined micronutrient models better predicted the level of CRP than the individual micronutrient models. Additionally, relationships strengthened when the analyses were limited to the individuals reporting chronic pain. In summary, findings highlight the complex relationships between anti-inflammatory micronutrients, inflammation, cellular aging, and chronic pain.

## Figures and Tables

**Table 1 nutrients-14-00266-t001:** Participant characteristics of people with and without chronic pain, *N* = 402.

Characteristics		*N* (%)
Sex	Women	255 (63.4)
	Men	147 (36.6)
Ethnicity/Race	Non-Hispanic-white	209 (52.0)
	Non-Hispanic-black	193 (48.0)
Education	≤High school	168 (41.8)
	>High school	234 (58.2)
Income ($)	0–19,999	121 (30.6)
	20,000–49,999	142 (35.9)
	50,000–79,999	64 (16.2)
	≥80,000	69 (17.4)
Study Site	University of Florida	251 (62.4)
	University of Alabama	151 (37.6)
Waist Hip Ratio (WHR) ^1^	Low	166 (41.3)
	Moderate	93 (23.1)
	High	143 (35.6)
Physical Activity/Week	<1/week	113 (28.4)
	1–3/week	181 (45.5)
	≥4/week	104 (26.1)
Tobacco Smoking Status	Never smoker	204 (51.3)
	Former smoker	110 (27.6)
	Current smoker	84 (21.1)
Number of Comorbidities	<1	186 (46.3)
	1	118 (29.4)
	≥2	98 (24.4)
Depressive symptoms ^2^	No depression	257 (63.9)
	Mild	75 (18.7)
	Moderate-severe	70 (17.4)
Chronic Pain Status	No chronic pain	82 (20.5)
	Chronic pain	318 (79.5)

Numbers and percentages presented are based on non-missing values of the overall study population. ^1^ WHR [55]: low (women < 0.80, men < 0.95), moderate (women 0.81–0.85, men 0.96–1.00), high (women > 0.86, men > 1.0). ^2^ Depressive symptoms (Center for Epidemiology Studies Depression score, CES-D score) [58]: no depression 0–9, mild 10–15, moderate-severe ≥ 16.

**Table 2 nutrients-14-00266-t002:** Associations between selected serum micronutrients and leukocyte telomere length in individuals with and without chronic pain.

	Unadjusted		Adjusted	
	*N*	β (95% CI)	*N*	β (95% CI)
Vitamin D	209	<−0.01 (<−0.01, <0.01)	206	<0.01 (<−0.01, <0.01)
Omega 6:3 ratio	127	−0.01 (−0.03, 0.01)	126	<−0.01 (−0.02, 0.02)
Combined model				
Vitamin D	127	<−0.01 (<−0.01, <0.01)	127	<0.01 (<−0.01, 0.01)
Omega 6:3 ratio	127	−0.01 (−0.03, 0.02)	127	<0.01 (−0.02, 0.02)

Regression estimates rounded to their nearest hundredth and <0.01 assigned for lower values; LTL was log transformed. Multivariable model adjustments: age, sex, race, study site, WHR, physical activity, and number of pain sites. Combined model: regression models included both vitamin D and omega 6:3 ratio as predictors along with other covariates.

**Table 3 nutrients-14-00266-t003:** Associations between selected serum micronutrients and C-reactive protein in individuals with and without chronic pain.

	Unadjusted		Adjusted	
	*N*	β (95% CI)	*N*	β (95% CI)
Vitamin D	211	−0.04 (−0.05, −0.02)	203	−0.02 (−0.04, <0.01)
Omega 6:3 ratio	163	0.10 (−0.01, 0.20)	157	0.11 (0.01, 0.21)
Combined model				
Vitamin D	163	−0.03 (−0.05, −0.01)	157	−0.01 (−0.03, 0.01)
Omega 6:3 ratio	163	0.06 (−0.05, 0.17)	157	0.10 (−0.01, 0.21)

Regression estimates rounded to their nearest hundredth and <0.01 assigned for lower values; CRP was log transformed. Multivariable model adjustments: age, sex, WHR, physical activity, tobacco smoking status, number of comorbidities, annual household income, and number of pain sites. Combined model: regression models included both vitamin D and omega 6:3 ratio as predictors along with other covariates.

## Data Availability

Data supporting the results were generated during the study and are not publicly available. Summary results and additional analyses requests related to this study can be accommodated on request from the corresponding author.

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
