# Peer review of "Associations between Vitamin D, Omega 6:Omega 3 Ratio, and Biomarkers of Aging in Individuals Living with and without Chronic Pain"

_nutrients, 2022, doi:10.3390/nu14020266_

Round 1

Reviewer 1 Report

Dear Authors

Thank you for exploring this important topic. The study is interesting because it is concerned the examination of inflammation,  cellular aging and pain in many configurations and relationships. In my opinion, this is very interesting manuscript that add many to our clinical knowledge. Although, taking into account the range of controlled parameters, I wonder whether the concentration of IL-6 should also be assessed in terms of its association with inflammation and pain.

Author Response

Thank you so much for your feedback. We appreciate it very much. In regards to your comment, we agree that the addition of IL-6 may also be helpful to evaluate relationships between the identified micronutrients and pain. For the current study, we were limited to CRP, a sentence has been added to the discussion section as a future directions suggestion (blue font).

“Regarding additional future directions, multiple time point measurements could improve interpretations of inflammation. For instance, multiple time point CRP measurements may provide a more accurate estimation of chronic systemic inflammation as opposed to using a single CRP measurement [76]. Although not specific to CRP, another study reported no statistically significant association between serum omega 3 and baseline LTL but found a significant positive association between serum omega 3 and change of LTL over time [40]. Even though some studies have shown cross-sectional associations between vitamin D, omega 3 and LTL [30, 42, 60], these examples indicate the possible relevance of using multiple time point measurements to assess both CRP and LTL measurements. Additionally, consideration of additional pro-inflammatory biomarkers such as IL-6 may also be informative.”

Reviewer 2 Report

The authors investigated the relationships between vitamin D, omega 6:3 ratio, CRP and LTL in chronic pain. Findings suggest a strong relationship between vitamin D, omega 6:3 ratio and CRP in people with chronic pain but not LTL.

The manuscript is well organized. Each section has been properly developed.

Only one remark. Please, add in the introduction section a part explaining the relationship between vitamin D, gut microbiota and chronic pain.

Author Response

Thank you so much for your feedback. We appreciate the suggested addition and understand this addition will improve our manuscript. Thus, the following sentences were added to the Introduction and Discussion (blue font):

Introduction:

 “Previous studies have suggested that individuals with chronic pain may benefit from consuming foods and micronutrients with anti-inflammatory properties [19-22]. However, the question remains if such foods and micronutrients are associated with longer LTL in people living with chronic pain. 25-hydroxyvitamin D (vitamin D) and omega 3 fatty acids seem to show some benefit toward reducing chronic pain [1, 5-8] and are associated with longer LTL [27-29]. Although previous studies indicate conflicting evidence of the association between vitamin D on LTL in the general population [30-32], two studies reported expression of LTL protective enzyme telomerase in individuals with chronic inflammatory conditions [28, 33]. Moreover, there is a developing body of evidence showing vitamin D deficiency is positively associated with both inflammation and chronic pain, which may be related to changes in pain signaling pathways (e.g., endocannabinoids system) [34] and gut microbial composition [34-36].  

Discussion (sub-topic 4.3: Biological Plausibility of Observed Associations):

 “Vitamin D is involved in the regulation and expression of pro-inflammatory mediators, inflammatory genes, and suppression of key inflammatory response pathways such as NF-κB [70, 71]. Possible relationships between vitamin D deficiency, pain signaling pathways (e.g., endocannabinoids system)[34], and altered gut microbial composition [34-36] have also been indicated. Similarly, omega 3 fatty acids are implicated in suppressing the inflammatory response including the suppression of NF-κB activity, reducing the impact of omega 6 driven pro-inflammatory cytokine production [72]. There may also be some overlap between the anti-inflammatory activity of these micronutrients and pain reduction [37, 73-75].”